# Effect of Fluoride Content of Mouthwashes on Superelastic Properties of NiTi Orthodontic Archwires

**DOI:** 10.3390/ma15196592

**Published:** 2022-09-22

**Authors:** Francisco Pastor, Juan Carlos Rodríguez, José María Barrera, José Angel Delgado García-Menocal, Aritza Brizuela, Andreu Puigdollers, Eduardo Espinar, Javier Gil

**Affiliations:** 1Departamento Ortodoncia, Facultad de Odontología, Universidad de Sevilla, Avicena s/n, 41009 Sevilla, Spain; 2Bioengineering Institute of Technology, Facultad de Medicina y Ciencias de la Salud, Universidad Internacional de Catalunya, Josep Trueta s/n, Sant Cugat del Vallés, 08195 Barcelona, Spain; 3Facultad de Odontología, Universidad Europea Miguel de Cervantes, C/del Padre Julio Chevalier 2, 47012 Valladolid, Spain; 4Departamento Ortodoncia, Facultad de Odontología, Universidad Internacional de Catalunya, Josep Trueta s/n, Sant Cugat del Vallés, 08195 Barcelona, Spain

**Keywords:** NiTi, orthodontic archwires, fluorides, superelasticity, transformation temperatures

## Abstract

The influence of sodium fluoride (NaF) concentration in mouthwashes on the properties of superelastic NiTi orthodontic wires has been studied. In this work, 55.8%Ni and 44.2%Ti (in weight) wires were introduced in commercial mouthwashes with different NaF contents (0, 130, 200 and 380 ppm). The release of Ni^2+^ and Ti^4+^ ions was by Inductively Coupled Plasma-Mass Spectrometry (ICP-MS) at 1, 4, 7 and 14 days. Superelastic orthodontic wires present at oral temperature the austenitic phase which is transformed into a plastic phase (martensite) by cooling. The temperatures at which this occurs are influenced by the chemical composition. The release of ions from the wire will produce variations in the temperatures and stresses of the stress-induced martensitic transformation. M_s_, M_f_, A_s_, A_f_ were determined by Differential Scanning Calorimeter (DSC). The transformation stresses (austenite to stress induce martensite) were determined with a servo-hydraulic testing machine at 37 °C. The surfaces for the different times and mouthwash were observed by Scanning Electron Microscope (SEM). The release of Ni^2+^ in mouthwashes with 380 ppm NaF concentrations reaches 230,000 ppb in 14 days and for Ti^4+^ 175,000 ppb. When NaF concentrations are lower than 200 ppm the release of Ni and Ti ions is around 1500 ppb after 14 days. This variation in compositions leads to variations in M_s_ from 27 °C to 43.5 °C in the case of higher NaF concentration. The increasing immersion time and NaF concentrations produce a decrease of Ni in the wires, increasing M_s_ which exceed 37 °C with the loss of superelasticity. In the same way, the stresses (tooth position corrective) decrease from 270 MPa to 0 MPa due to the martensitic phase. The degradation can produce the growth of precipitates rich in Ti (Ti_2_Ni). These results are of great interest in the orthodontic clinic in order to avoid the loss of the therapeutic properties of superelastic NiTi due to long immersion in fluoride mouthwashes.

## 1. Introduction

The use of fluoride-containing mouthwashes prevents caries due to the formation of fluorapatite in the mineral content of the teeth which is much more resistant to caries, and also has a beneficial action in the treatment of dental hypersensitivity [1,2,3,4]. This makes the use of gels and mouthwashes common during orthodontic treatment. However, the concentration of fluorides in aqueous or alcoholic solutions promotes the reaction of fluorides with the metals that form orthodontic wires, especially Ti, causing the release of metal ions into the environment [5,6,7,8]. Mouthwashes have different concentrations of sodium fluorides which in physiological media form HF as all sodium salts are soluble in aqueous media. Nakagawa has already observed electrochemical corrosion at 30 ppm HF, with these concentrations being capable of destroying the passivation layers of orthodontic wires [9,10,11].

Many of the metals and alloys used as orthodontic wires have a passivation layer, in the case of Ti or beta-Ti or NiTi wires a Ti oxide layer of 2 to 200 nm is formed, in the case of stainless steels a chromium oxide layer is formed to protect the metal from external fatty agents. Fluoride ions have the ability to dissolve this layer with the formation of fluorides that reduce corrosion resistance and release metal ions into the physiological environment [12,13].

In orthodontics, the destruction of the passivation layer is more serious as the friction of the wire with the bracket causes an acceleration of the ion release process and favors electrochemical corrosion [14,15]. These ions released into the environment can lead to an inflammatory response in the surrounding tissues in the case of Ti dental implants and more localized cases are observed. In the case of orthodontics, the ions dissolve in the salivary medium and can therefore be transported to the rest of the human body. This ionic diffusion can lead to cytotoxicity, mutagenicity and/or carcinogenic reactions [16,17,18,19].

Attempts have been made to improve the strength of the passivation layer of orthodontic wires containing Ti such as nitriding formed Ti nitrides which improves the sliding between the wire and the bracket and increases the passivation of the layer or the incorporation of 0.1 to 0.3% Pt or Pd in the composition of the orthodontic wire which significantly improves the stability of the inert Ti oxide layer up to mouthwashes containing 2% NaF [20,21,22,23].

Studies have been carried out on Ti dental implants on the effect of fluoride solutions in mouthwashes but there are no in-depth studies on orthodontic wires, not only on ion releases but also how it affects the properties of the wires, especially NiTi wires. These superelastic properties can vary with heat treatment [24,25], grain size increase [26,27] or surface modification of their chemical composition [28]. One factor that may affect the chemical composition is the release of Ti and Ni ions due to fluoride-containing mouthwashes.

The objective of our research is to determine how the NaF concentration in mouthwashes influences the release of Ni and Ti ions in NiTi superelastic wires. It is well known that variations in the chemical compositions of orthodontic wires have a major influence on the transformation temperatures and stresses and will therefore affect the functionality of the wires. This study has not been carried out and at the various international orthodontic congresses this need has been raised for the knowledge of clinicians.

## 2. Materials and Methods

In this case, 45 NiTi commercial archwires (Neo Sentalloy 0.32 mm diameter, GAC, West Columbia, SC, USA) were used in this study. The chemical composition of the NiTi was in weight percentage: Ni: 55.8%, Ti: 44.2%. These percentages compositions were determined by means dispersive energy of X-ray. Samples measured 0.46 mm in diameter and 45 mm in length were cut from the original archwires.

The wires were placed in the mouthwashes with the same chemical composition except for the sodium fluoride content at a constant temperature of 37 °C. The sodium fluoride contents can be seen in Table 1. Commercial mouthwashes can be classified as low sodium fluoride content up to approximately 200 ppm and high content with a high bactericidal capacity from 300 ppm onwards. These studies have been carried out on Ti dental implants and a sharp change in Ti ion concentration is observed between the low and high contents.

### 2.1. Ion Release

The ion release test was performed immersing the archwires in 6 mL of the three different mouthwashes (Table 1) and 37 °C, for 1, 4, 7 and 14 days. The immersion times were obtained from the conclusions of the several health authorities and exposed in the International Orthodontic Conference [29,30,31,32], where orthodontic clinicians noted that in some cases, NiTi wires and other NiTi devices were introduced at sleep times in mouthwashes with different concentrations of sodium fluoride [33]. Treatments could in some cases be as long as 25 days and it was these times that could give insight into the behaviour of NiTi wires [33,34]. Ion-release quantification was carried out by inductively coupled plasma-mass spectrometry (ICP-MS) by using Perkin Elmer Optima 320RL equipment (Waltham, MA, USA).

### 2.2. Calorimetric Tests

Superelasticity is possible due to the elastic characteristics of the martensitic transformation (austenite → martensite), in the case of steels the martensitic transformation presents plastic characteristics and consequently the transformation is not reversible. For the alloys with thermoelastic transformation this can be produced by cooling or when the stress is applied [35,36]. When a stress is loaded to the austenitic phase above its A_f_ temperature, an elastic martensite is stress induced. That is, the deformed material reverts to its original shape when the stress is released. The stress necessary to grow stress induced martensite (SIM) is a linear function of temperature. The stress σ^SIM^^→AUS^ increases with increasing temperature while the yield stress of the austenitic phase decreases with increasing temperature [37,38]. Transformation hysteresis (i.e., the difference between σ^AUS^^→SIM^ and σ^SIM^^→AUS^) indicates the quantity of dispersive energy taking place during the growth SIM [39,40,41,42].

Five samples for each treatment were analyzed (45 mm long, 0.46 mm diameter). The transformation temperatures were measured by Differential Scanning calorimeter (Mettler Toledo S 10, Columbus, OH, USA) [35,36,37]. M_s_ and A_s_ transformation temperatures occur when there is a sudden increment in the calorimetric signal. In the same way, the final temperatures, M_f_ and A_f_, were determined as when the calorimetric signal returned to the base line [37].

### 2.3. Mechanical Tests

The results of stress-induced martensite were determined by tensile tests by servo-hydraulic testing machine (MTS-Bionix 858, Minneapolis, MN, USA), working at a cross-head speed of 10 mm/min in artificial saliva at 37 °C [22,23,43,44]. Five cylinders of 0.457 mm in diameter and of 45 mm in length were tested for each treatment.

### 2.4. Electronic Microcopy

The surfaces of the samples were observed using a SEM (JEOL 1200 EXII Microscopy Tokyo, Japan equipped with a link LZ5 EDS (Jeol, Tokyo, Japan), which was also used for determining the chemical composition.

Transmission Electron Microscopy (TEM) (Jeol 4200, Tokyo, Japan) samples was used to observe the precipitates on the matrix. Samples were ground with 1200 grain size paper to a thickness of 150 μm and electrochemically polished using a double jet thinning technique. The chemical etching was realized by 35% Perchloric Acid, 35% Methanol and 40% 2-Butoxietanol electrolyte with 10 Volts of tension and electrolyte temperature of −15 °C.

### 2.5. Statistical Analysis

The data was statistically analyzed using Student’s *t*-tests, one-way ANOVA tables and Turkey’s multiple comparison tests in order to evaluate any statistically significant differences between the sample groups. The differences were considered significant when *p* < 0.05. All statistical analyses were performed with MinitabTM software (Minitab release 13.0, Minitab Inc., State College, PA, USA).

## 3. Results

The results of ion release at different immersion times in mouthwashes with different sodium fluoride contents can be seen for Ni in Figure 1 and for Ti ions in Figure 2.

In Figure 3 we can see the original structure of the NiTi wires, where we can see the austenitic structure that gives the superelasticity to the wires.

Figure 4 shows the NiTi wire structures after immersion in the different mouthwashes for 4 and 14 days. The microstructures show the appearance of martensitic plates (M) in an austenite matrix (A).

It can be seen that with increasing immersion time there is an increase in the star-shaped acicular structures characteristic of the martensitic structure. It can be seen how with the mouthwash 3, which contains more NaF, the martensitic phase is much more abundant and in this microstructure after 14 days of immersion it is totally martensitic without observing the austenitic phase and therefore there is a loss of its superelasticity.

By SEM image analysis, the percentage of austenitic and martensitic phases in orthodontic wire at different immersion times has been determined for each mouthwash. The results are shown in Table 2.

In the samples subjected to immersion with 380 ppm NaF for 14 days, we were able to observe the appearance of precipitates in the matrix by TEM. The white precipitates were distributed throughout the martensitic structure and were not quantifiable for the image analysis system, although they could be observed at high resolution. To determine the nature and characteristics of these precipitates, we left the samples immersed for 28 days in the NaF-rich solution and we could see the increase in number and size of the precipitates, as shown in Figure 5. Table 2. Determination of the phase’s percentages in the different NiTi archwires treated with different mouthwashes at different immersion times.

Table 3 shows the transformation temperatures of NiTi treated with the different mouthwashes at different immersion times. These results indicate the temperatures where the martensitic transformation starts (M_s_) at which temperature is 100% martensite phase (M_f_) and the values of the reverse martensite to austenite transformation (A_s_ and A_f_). As is well known, it is the austenitic phase that exhibits the superelastic behavior which is of great effectiveness for tooth movement in orthodontic techniques.

Table 4 shows the values of the transformation stresses from austenite to martensite and of the retransformation from stress-induced martensite to the austenitic phase.

## 4. Discussion

As can be seen, the release of both Ti^4+^ and Ni^2+^ions are sensitive to the sodium fluoride content of the mouthwash. This fluoride, as has been described in different investigations [45,46,47] is the cause of the release of ions into the medium. It has been observed that there is a change in the release of ions when we exceed 200 ppm of sodium fluoride. Furthermore, it can be observed that the solution containing 380 ppm does not produce a stabilization in the release of ions over time, as is the case with most of the metals used to make orthodontic wires, but rather an almost linear growth [26,27,28,45,46,47].

This ion release is more pronounced in Ni than in Ti, although in both cases it is very high, exceeding 200.000 ppb after 14 days of immersion. Consequently, the chemical compositions of orthodontic wires are changing significantly. As the Ni^2+^ ion leaching becomes more pronounced, the wire becomes richer in Ti.

As is well known, NiTi wires are very sensitive to chemical composition since small variations will produce changes in transformation temperatures [26,27,28], as demonstrated by the results in Table 3. The original wire presents a M_s_ of 27 °C which does not indicate that the wire when placed in the mouth at 37 °C presents a single austenitic phase. This phase is the desired one in orthodontics as the deformation induced in the wire by the dental malposition, the wire due to the superelastic property will exert tension to return the wire to its ideal arch and correct the dental occlusion.

As can be seen from the results, Ti enrichment causes an increase in the M_s_ temperature. In the case of mouthwashes 1 and 2, this increase in M_s_ temperatures is smaller than in mouthwash 3, where the loss of Ni is more important. This fact causes the proportions of the austenitic and martensitic phases to vary, and the martensite increases its proportion in the wire as the action of the fluoride causes more Ni to be lost. In the case of mouthwash 3, we can see that the loss of Ni is so important that after 4 days the wire has lost the superelastic effect due to the decrease of the austenitic phase in the microstructure.

Teeth movement in orthodontic treatment is achieved by applying loads to dental pieces which produce a bone remodeling processes. The elastic strain of an orthodontic wire and the release of its elastic energy over a time provokes rise to the correcting loads. The ion release provokes the variation of these properties producing decrease of correcting forces until de no functionality archwires. It is well known that the optimal teeth movement is obtained when the loads applied are low and continuous in the period of orthodontic treatment. These loads (low and continuous) decrease the tissue loss and provoke a relatively constant stress in the periodontal ligament during tooth movement [24,25,26].

The transformation stress results are important from a clinical point of view as the retransformation stresses are the stresses exerted by the orthodontic wire on the tissues to cause tooth movement. As can be seen from the results, the effect of sodium fluoride causes a change in the chemical compositions leading to a change in the transformation and retransformation temperatures (Table 3) and consequently in the transformation and retransformation stresses (Table 4). In the case of samples with zero or very low retransformation stresses, the orthodontic wire can be considered as no longer active and does not produce tooth correction.

From the results in Table 3 we can determine the thermal hysteresis of the transformation, which is defined as A_s_-M_f_ and gives the estimate of the energy dissipated in the process. It can be seen in Table 5 how the hysteresis values increase with the immersion time. This fact can be justified since NiTi is leaching ions from the wire surface and therefore there is a solid-state diffusion that makes the system lose energy, as has been studied by some authors in other materials with signature memory such as CuZnAl and other copper-based alloys with thermoelastic martensitic transformation [43,44].

The temperature increases M_s_ due to Ti enrichment in the wire cause the transformation and retransformation stresses to decrease as the stress-induced martensitic transformation needs less stress to be obtained (Table 4). The thermodynamic jump is smaller as the temperature is closer to the human body temperature of 37 °C. Transformation temperatures above 37 °C indicate that the whole structure is martensitic and therefore no stress is needed to induce martensite. The wires are totally passive; they do not exert any force on the dental tissues.

The stresses for orthodontic therapy are the retransformation stresses as this is the stress that the wire will apply to the teeth. These stresses are constant and must be low to avoid hyalinization of the tissues. Clinicians should also be aware of how mouthwash treatment times vary the corrective tensions. For this reason, it is advisable to warn patients treated with NiTi wires not to leave the wires in mouthwashes for a long time in immersion. Deviations have been observed in orthodontic treatment when patients leave their wires in mouthwash at bedtime.

As can been observed by SEM (Figure 5) when the time in immersion is long, the chemical composition can produce a stoichiometric precipitate rich in Ti. These precipitates have been observed by TEM (Figure 6) and they present an elliptical shape with an average diameter of 490 + 203 nm depending on the immersion time. Electron diffraction technique was used for determining their structure. The results showed a FCC crystallographic structure with lattice parameter of 11.278 Å. which was associated with a Ti_2_Ni precipitates. The martensitic plate size is smaller than in equiatomic material due to this material presented a lesser amount of free-precipitates space. This means that the precipitates are obstacles which impede the martensitic plate growth and change their direction [22,23,24,25].

It was not possible to compare our results on the relationship between transformation temperatures and voltages with immersion times in mouthwashes with different concentrations of sodium fluoride as there is no published work on this aspect. What we have been able to confirm is the relationship of how transformation temperatures and stresses vary with variations in the chemical composition of orthodontic wires, confirming how decreases in Ni lead to increases in transformation temperatures and therefore decreases in stress-induced martensitic transformation stresses. We have also been able to verify the appearance of Ti-rich precipitates of the same morphology and stoichiometry as those obtained in this study.

In addition to the chemical composition, grain sizes can also influence the transformation stresses. The smaller the crystal size, the higher the transformation temperatures and therefore the lower the transformation stresses. However, there will be no grain size variations in our wires as they will not be subjected to heat treatment or welding processes.

## 5. Conclusions

The effect of sodium fluoride from mouthwashes on NiTi wires causes a very important release of Ni and Ti ions. The amount of Ni ions released is higher than that of Ti reaching values around 220,000 ppb and 180,000 ppb for Ni and Ti, respectively, when the wire is immersed in 380 ppm of NaF for 14 days. Ion release values for mouthwashes with less than 200 ppm are around 1500 ppb for Ni and Ti. This confirms the results of the abrupt change of NaF concentration behavior on ion release already observed for dental implants and prosthetic metals. This variation of the chemical composition causing an impoverishment of the Ni increases the temperatures from 27 °C to temperatures higher than that of the human body (37 °C) up to values higher than 43 °C, which causes that the wires can present martensitic structure and therefore the wire is totally inactive. The reduction of the austenitic phase in orthodontic wires causes a reduction in the corrective stresses on the teeth, which makes them passive, without superelastic behavior. The increase of Ti in the wire even produces Ti_2_Ni precipitates. The presence of Ti_2_Ni produces variations in the contents of Ni and Ti of the archwire. The formation of martensite occurs, causing the loss of superelastic behavior. The results should help clinicians to prevent patients from leaving their orthodontic wires in sodium fluoride-containing mouthwashes for a long time.

## Figures and Tables

**Figure 1 materials-15-06592-f001:**
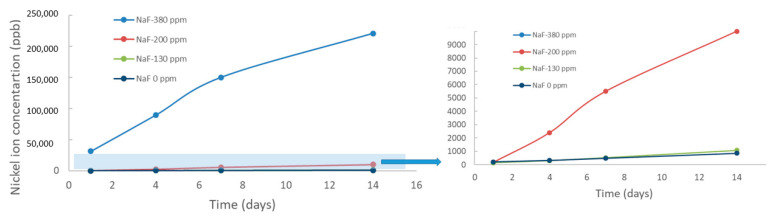
Ni ion release in ppb at different immersion times for each mouthwash studied. The shaded area is enlarged for the 0, 130 and 200 ppm NaF concentrations.

**Figure 2 materials-15-06592-f002:**
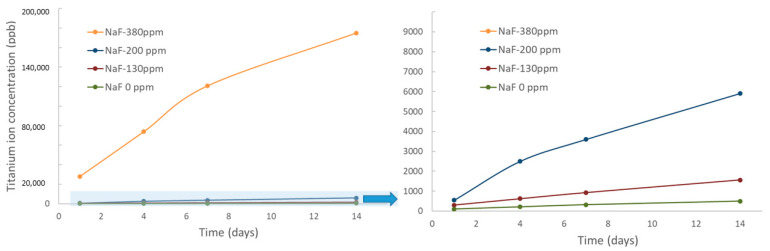
Ti ion release in ppb at different immersion times for each mouthwash studied. The shaded area is enlarged for the 0, 130 and 200 ppm NaF concentrations.

**Figure 3 materials-15-06592-f003:**
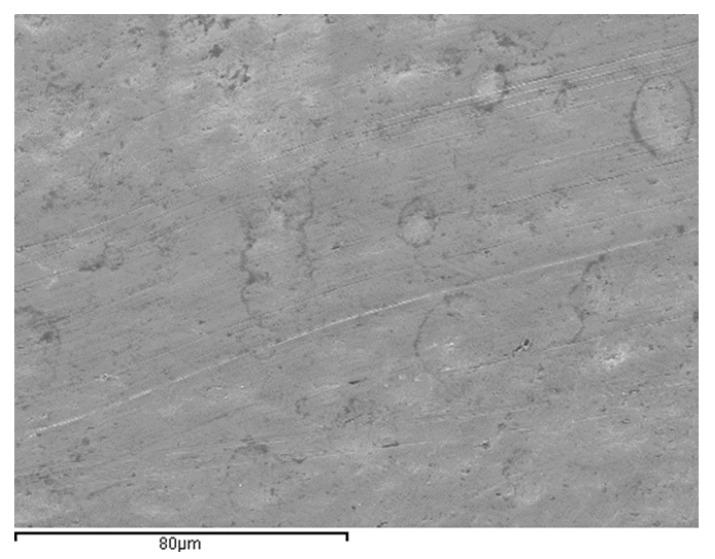
As-received NiTi orthodontic wire. The microstructure is completely austenite.

**Figure 4 materials-15-06592-f004:**
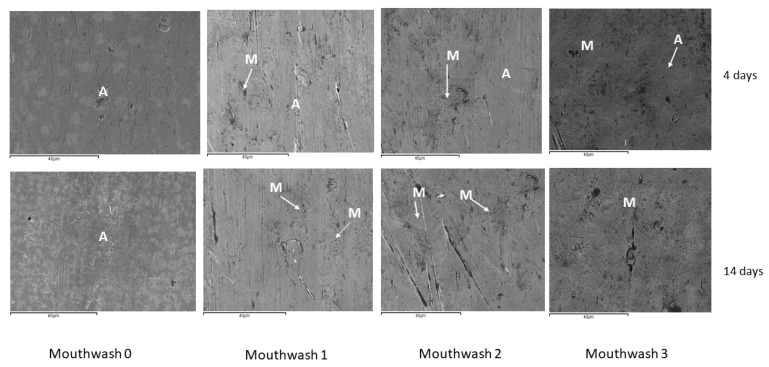
Microstructures of NiTi archwires after immersion for 4 and 14 days for the different mouthwashes. In the figures have been marked the phases martensitic (M) and austeinitic (A). The arrows indicate the phases.

**Figure 5 materials-15-06592-f005:**
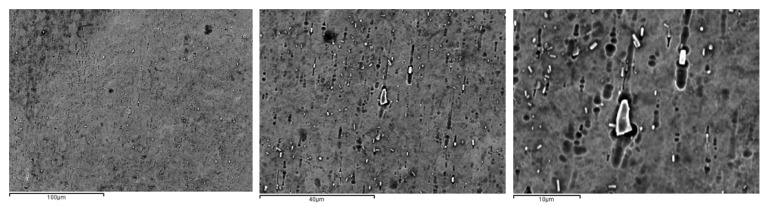
Precipitates formed in NiTi archwire after 28 days in NaF solution (380 ppm). The images have different magnifications.

**Figure 6 materials-15-06592-f006:**
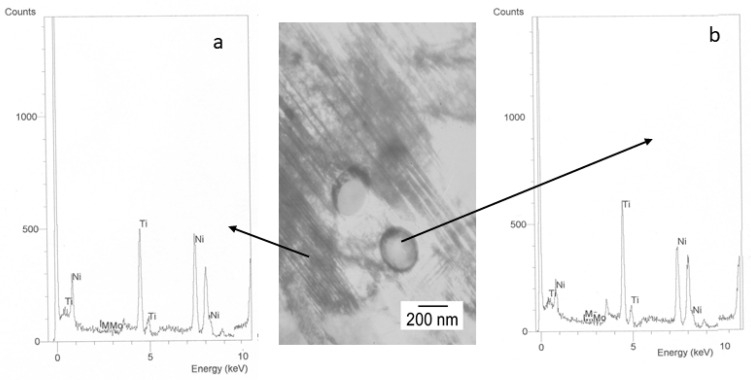
Ti_2_Ni precipitates in martensite observed by TEM. Zone axis [110]. EDS for matrix (**a**) and precipitates (**b**).

**Table 1 materials-15-06592-t001:** NaF composition of the different mouthwashes.

Mouthwahes	NaF (ppm)
0	0
1	130
2	200
3	380

**Table 2 materials-15-06592-t002:** Percentages of the austenite and martensite phases in orthodontic wires for each sodium fluoride content.

Mouthwash	Time (Days)	%Austenite	%Martensite
Original	0	100	0
0	1	100	0
0	4	100	0
0	7	100	0
0	14	100	0
1	1	100	0
1	4	98	2
1	7	87	13
1	14	80	20
2	1	100	0
2	4	94	6
2	7	78	22
2	14	61	39
3	1	81	19
3	4	43	57
3	7	0	100
3	14	0	100

**Table 3 materials-15-06592-t003:** Transformation temperatures (°C) for the Ni-Ti archwires studied with the different immersion times for each mouthwash.

Mouthwash	Time (Days)	M_s_	M_f_	A_s_	A_f_
Original	0	27.2 ± 0.3	16.1 ± 0.4	20.0 ± 0.1	32.3 ± 0.7
0	1	27.0 ± 0.5	16.9 ± 0.9	22.0 ± 0.9	34.5± 0.5
0	4	26.9 ± 0.4	16.0 ± 0.5	21.0 ± 0.5	32.0 ± 0.9
0	7	27.2 ± 0.3	16.1 ± 0.4	20.0 ± 0.1	32.3 ± 0.8
0	14	27.3 ± 0.3	16.0 ± 0.4	20.1 ± 0.1	32.3 ± 0.7
1	1	27.9 ± 0.6	16.1 ± 0.5	21.0 ± 0.1	40.3 ± 0.7
1	4	27.8 ± 0.3	15.8 ± 1.0	21.3 ± 0.9	40.4 ± 0.5
1	7	30.3 ± 0.2	9.2 ± 0.3	17.1 ± 0.4	48.4 ± 0.5
1	14	37.4 ± 1.2	17.4 ± 1.3	28.1 ± 1.9	57.6 ± 1.2
2	1	30.6 ± 2.2	10.3 ± 0.6	14.5 ± 1.4	38.4 ± 0.9
2	4	33.6 ± 0.3	1.4 ± 0.3	16.1 ± 0.4	38.1 ± 0.6
2	7	36.6 ± 1.3	9.4 ± 1.2	19.1 ± 0.9	39.3 ± 1.2
2	14	38.9 ± 0.9	10.0 ± 0.8	24.2 ± 2.3	47.9 ± 0.9
3	1	32.4 ± 0.4	13.4 ± 0.1	27.3 ± 0.5	36.2 ± 0.9
3	4	36.4 ± 0.4	14.4 ± 0.3	34.2 ± 0.5	47.3 ±1,0
3	7	39.7 ± 1.4	16.4 ± 0.1	43.2 ± 0.5	56.2 ± 2.2
3	14	45.4 ± 0.9	19.6 ± 1.1	54.1 ± 1.4	72.3 ± 2.7

**Table 4 materials-15-06592-t004:** Critical stresses at different test times for each mouthwashes studied.

Mouthwash	Time (Days)	σ^β^^→SIM^ (MPa)	σ^SIM^^→β^ (MPa)
Original	0	270 ± 15	151 ± 19
0	1	276 ± 10	141 ± 20
0	4	260 ± 14	138 ± 18
0	7	257 ± 16	155 ± 14
0	14	278 ± 19	150 ± 16
1	1	279 ± 20	143 ± 17
1	4	278 ± 13	140 ± 15
1	7	180 ± 20	88 ± 10
1	14	137 ± 12	72 ± 10
2	1	230 ± 22	130 ± 22
2	4	198 ± 13	98 ± 16
2	7	60 ± 15	29 ± 10
2	14	12 ± 9	4 ± 2
3	1	82 ± 23	26 ± 9
3	4	13 ± 6	4 ± 3
3	7	0	0
3	14	0	0

**Table 5 materials-15-06592-t005:** Thermal hysteresis of the NiTi treated with different mouthwashes at different times.

Mouthwash	Time (Days)	Hysteresis
Original	0	3.9
0	1	5.0
0	4	5.1
0	7	4.9
0	14	4.9
1	1	4.9
1	4	6.5
1	7	7.9
1	14	10.7
2	1	4.2
2	4	14.7
2	7	9.7
2	14	14.2
3	1	13.9
3	4	19.8
3	7	26.8
3	14	34.5

## Data Availability

The authors can provide details of the research requesting by letter and commenting on their needs.

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
