# Peer review of "Effect of Fluoride Content of Mouthwashes on Superelastic Properties of NiTi Orthodontic Archwires"

_materials, 2022, doi:10.3390/ma15196592_

Round 1

Reviewer 1 Report

In the present work, the authors study the effect of fluoride on the superelastic properties of NiTi. The authors introduced a superelastic NiTi orthodontic wire in three commercially available mouthwashes with different sodium fluoride contents and studed the release of nickel and titanium ions by Inductively Coupled Plasma-Mass Spectrometry. The results showed a release of nickel ions higher than that of titanium with immersion times. And these variations of the chemical composition cause a very significant decrease of the austenitic phase and the transformation stresses also decrease. These results are great interest in the orthodontic clinic in order to avoid the loss of the therapeutic properties of superelastic NiTi.

Figure 1& Figure 2: all of the data should be repeated at leat three times, but the present work just show one times data. It is very important to show a data there is no NaF present. It looks NaF-380ppm has huge effect on Titanium ion comparing to NaF-200ppm. It is better to add more [NaF] between NaF-380ppm and NaF-200ppm.

Figure 3&Figure 4, need to show the image of NaF-0ppm.

Author Response

Dear Reviewer,

Thanks for taking the time to review our manuscript and suggest to us to improve our work by providing a lot more detail. We have done so, and we are now submitting a manuscript that not only addresses the points the you specifically raised but also many others that we have considered in order to deliver what we think is a much improved version of our work. This version includes more paragraphs, English grammar revisions in all main sections, new references. Thanks a lot. We are looking forward to your comments.

Sincerely,

Francisco-Javier Gil Mur

Reviewer 1

In the present work, the authors study the effect of fluoride on the superelastic properties of NiTi. The authors introduced a superelastic NiTi orthodontic wire in three commercially available mouthwashes with different sodium fluoride contents and studed the release of nickel and titanium ions by Inductively Coupled Plasma-Mass Spectrometry. The results showed a release of nickel ions higher than that of titanium with immersion times. And these variations of the chemical composition cause a very significant decrease of the austenitic phase and the transformation stresses also decrease. These results are great interest in the orthodontic clinic in order to avoid the loss of the therapeutic properties of superelastic NiTi.

  1. Figure 1& Figure 2: all of the data should be repeated at leat three times, but the present work just show one times data. It is very important to show a data there is no NaF present. It looks NaF-380ppm has huge effect on Titanium ion comparing to NaF-200ppm. It is better to add more [NaF] between NaF-380ppm and NaF-200ppm.

In Figures 1 and 2 the values of a 0 ppm mouthwash solution have been entered. The values for the 0, 130 and 200 ppm NaF mouthwashes have been enlarged in order to have a better resolution of the values due to the large differences with the 380 ppm mouthwashes. The reviewer's comment is very timely and I believe it improves the manuscript and we thank him for it.

This is due to the fact that the mouthwashes are divided into soft and hard ones due to the concentration of NaF. At 325-350 ppm of NaF there is a radical change in the release of ions, as observed in the experimentation on titanium dental implants and which has been referenced in the text. The abrupt change that occurs is presumably due to the limiting concentration above 325 ppm NaF and therefore we would not see a gradual change of the same.

  1. Figure 3&Figure 4, need to show the image of NaF-0ppm.

In accordance with the reviewer's comment, we have changed the figure by adding the value of 0 ppm NaF.

Reviewer 2 Report

Authors present a study on the importance of time for contact with mouthwash solutions and Ni-Ti alloys used as archwires. Despite the idea is nice, the work is not applicable since the time used for analysis of released ions or even dissolution of the mettalic surface is too long, totally out of good sense.

Authors should try to see all the analysis in a very short time, the one which is meaning for the actual use of mouthwashes for consumers.

Author Response

Reviewer 2

Authors present a study on the importance of time for contact with mouthwash solutions and Ni-Ti alloys used as archwires. Despite the idea is nice, the work is not applicable since the time used for analysis of released ions or even dissolution of the mettalic surface is too long, totally out of good sense.

Dear reviewer:

Thanks for taking the time to review our manuscript and suggest to us to improve our work by providing the comment.

Authors should try to see all the analysis in a very short time, the one which is meaning for the actual use of mouthwashes for consumers.

Regarding the reviewer's comment that the times should be shorter and that it is of no interest, I would like to say that a commission of the European Society of Orthodontics, as well as the SEDO and the European Union authorities suggested that we carry out this study under the conditions proposed. Several references have been introduced in the text.  The fluorides in the fluoride collectors affect the NiTi wires with a high ion release and affect the orthodontic properties of the NiTi wires in a very important way. At the meetings, the problem of an increasing number of patients who leave their orthodontic wires immersed in mouthwash solutions containing sodium fluorides overnight (at bedtime) was raised. Orthodontic treatments are approximately 2 to 4 weeks and it is for this reason that these times are planned.

Likewise, the compositions of the most commercially available mouthwashes were established, which have fluoride contents ranging from 100 ppm to 380 ppm, in this case the mouthwashes known as "Hard", which are very damaging to the wires. These studies, which are intended to be published in the journal Materials and others that are being carried out under the same conditions in other fields cytocompatibility, accumulation of titanium ions in organs, will serve to establish a criterion that would be established by the European Union and studied with other FDA health agencies, etc. to explain to clinicians and patients the problem that this immersion in sodium fluoride solutions can cause for a long period of time.

The time of 14 days is very long but it was decided to study it to give an idea of the degradation of the NiTi orthodontic wire.

This aspect, for which we thank the reviewer for his comment, has been introduced in the introduction to explain to the readers the reasons for the times and concentrations of the mouthwashes.

We have done so, and we are now submitting a manuscript that not only addresses the points the you specifically raised but also many others that we have considered in order to deliver what we think is a much improved version of our work. This version includes more paragraphs, English grammar revisions in all main sections, new references.

Thanks a lot. We are looking forward to your comments.

Sincerely,

Francisco-Javier Gil Mur

Reviewer 3 Report

The abstract shall specify the SUMMARY of the article with numbers rather than general discussion.

Nevertheless, the introduction fails the present the state of knowledge in the literature regarding the subject and study area of this manuscript and fails to identify a gap that this manuscript intends to contribute. In this regard, please follow the following steps: (a) provide background information and set the context, (b) introduce the specific topic of your research and explain why it is important, (c) mention past attempts to solve the research problem or to answer the research question and d) conclude the Introduction by mentioning the specific objectives of your research. Additionally, at the end of the introduction, the authors described the purpose (objectives) of the study, mentioning also the methods used, obviously briefly, but this should be explained in more detail in the methods.

Materials and methods section is of low quality. Example: The NiTi archwires used are prepared in your laboratory or commercial samples? The tools and experimental devices and methods should be described similarly.

Results and discussion section must be considerably improved/ more technically presented. Please apply. All the results obtained should be compared with the reports (preferably recent) in the literature. More care should be taken to present the results and facilitate understanding of the work.

The abstract (again) and conclusions sections are poor and needs to be rewritten for better understanding. They must be extended, by giving some specific results of your research.

Specific comments: How did you choose the release time intervals (1, 4, 7 and 14 days)? Please replace “,”by “.”when referring to numbers.

Finally, I consider that the paper is not proper for publication in the present format and must be "Reject". Nevertheless, the efforts of performing all the experiments have been significant and I hope that in the near future all the issues will be solved.

Author Response

Reviewer 3

Dear Reviewer,

Thanks for taking the time to review our manuscript and suggest to us to improve our work by providing a lot more detail. We have done so, and we are now submitting a manuscript that not only addresses the points the you specifically raised but also many others that we have considered in order to deliver what we think is a much improved version of our work. This version includes more paragraphs, English grammar revisions in all main sections, new references. Thanks a lot. We are looking forward to your comments.

Sincerely,

Francisco-Javier Gil Mur

  1. The abstract shall specify the SUMMARY of the article with numbers rather than general discussion.

The abstract has been modified according to the reviewer.

  1. Nevertheless, the introduction fails the present the state of knowledge in the literature regarding the subject and study area of this manuscript and fails to identify a gap that this manuscript intends to contribute. In this regard, please follow the following steps: (a) provide background information and set the context,

Done

  1. (b) introduce the specific topic of your research and explain why it is important,

Done

  1. (c) mention past attempts to solve the research problem or to answer the research question and

Done

  1. d) conclude the Introduction by mentioning the specific objectives of your research. Additionally, at the end of the introduction, the authors described the purpose (objectives) of the study, mentioning also the methods used, obviously briefly, but this should be explained in more detail in the methods.

Done

  1. Materials and methods section is of low quality. Example: The NiTi archwires used are prepared in your laboratory or commercial samples? The tools and experimental devices and methods should be described similarly.

According to the reviewer, it has been introduced in the text that the orthodontic wire is commercial, giving the data and characteristics of the wire. The material and methods part has been revised.

  1. Results and discussion section must be considerably improved/ more technically presented. Please apply. All the results obtained should be compared with the reports (preferably recent) in the literature. More care should be taken to present the results and facilitate understanding of the work.

In agreement with the reviewer we have improved the results and discussion. We have not found any work on the influence of sodium fluoride content on transformation temperatures and stresses. However, we have commented on the research on how variations in chemical compositions influence the transformation temperatures and stresses that exert the orthodontic corrective function. We have also been able to confirm the presence of titanium-rich precipitates and the morphology of the precipitates when the alloy becomes titanium-rich. I believe that thanks to the reviewer's comment, the discussion has been improved.

  1. The abstract (again) and conclusions sections are poor and needs to be rewritten for better understanding. They must be extended, by giving some specific results of your research.

The abstract and conclusions have been improved following the comment of the reviewer. Results have been introduced.

  1. Specific comments: How did you choose the release time intervals (1, 4, 7 and 14 days)? Please replace “,”by “.”when referring to numbers.

The reasons for studying these immersion times have been explained and the reviewer's commentary on the text has been taken into account.

Finally, I consider that the paper is not proper for publication in the present format and must be "Reject". Nevertheless, the efforts of performing all the experiments have been significant and I hope that in the near future all the issues will be solved.

The authors hope that the corrections suggested by this reviewer and the improvements of the minor revisions of the other reviewers can be considered for publication. New texts, new references, new tests to confirm and justification of results and values of materials and methods have been introduced following the times and concentrations suggested by the EU health agenda.

Round 2

Reviewer 1 Report

All my concerns have been resolved.

May need to check English I never check them carefully. And also figures, they are not very clear. Thanks. 

Author Response

Reviewer 1.

All my concerns have been resolved.

May need to check English I never check them carefully. And also figures, they are not very clear. Thanks. 

Thank you for your comments. English and figures have been improved.

Reviewer 2 Report

Authors have changed almost all the manuscript according the referees suggestions, and then the manuscript is ready for publication.

Author Response

Reviewer 2

Authors have changed almost all the manuscript according the referees suggestions, and then the manuscript is ready for publication.

Thank you very much.

Reviewer 3 Report

Based on this revised version of the manuscript, unfortunately there are still unanswered questions and weaknesses in this paper.

The abstract is a shortened version of the manuscript and should entice readers to read the full text by highlighting the most important findings. Keeping the abstract concise while covering the important information in an attractive mode requires careful writing and revision. Please shorten the abstract section accordingly.

Introduction. Too many references for a sentence make no sense (lines 83, 86-88). Please add more relevant (for the manuscript) information.

The Results section objectively reports what you found, without speculating on why you found these results, while the Discussion section interprets the meaning of the results, puts them in context and explains why they matter. Please apply.

Please replace “,”by “.”when referring to numbers.

Please use chemical symbols, i.e. nickel - Ni2+, scanning electron microscope – SEM, etc.

Author Response

REVIEWER 3

Based on this revised version of the manuscript, unfortunately there are still unanswered questions and weaknesses in this paper.

  1. The abstract is a shortened version of the manuscript and should entice readers to read the full text by highlighting the most important findings. Keeping the abstract concise while covering the important information in an attractive mode requires careful writing and revision. Please shorten the abstract section accordingly.

In accordance with the reviewer's comment, the abstract has been shortened in line with the guidelines of the authors of the journal Materials. Thank you very much for your comment which improves the quality of the paper.

  1. Too many references for a sentence make no sense (lines 83, 86-88). Please add more relevant (for the manuscript) information.

Following the reviewer's comment, we have reduced 5 references from these paragraphs and left the most significant ones.

  1. The Results section objectively reports what you found, without speculating on why you found these results, while the Discussion section interprets the meaning of the results, puts them in context and explains why they matter. Please apply.

Thank you very much for your comment. The authors have selected the paragraphs commenting on the results and have injected them into the discussion part as suggested by the reviewer.

  1. Please replace “,”by “.”when referring to numbers.

All decimals have been changed the , by the . However, according to the journal's guidelines the thousands from 10,000 onwards are left with the decimal point.

  1. Please use chemical symbols, i.e. nickel - Ni2+, scanning electron microscope – SEM, etc.

Done. Thank you